# Pulsed Electric Fields vs. Pectolytic Enzymes in Arinto Vinification: Effects on Yield and Oenological Parameters

Mafalda Aguiar-Macedo [1], Luis M. Redondo [2],*, Marcos Teotónio Pereira [1] and Carlos Silva [3]

1   EnergyPulse Systems, EPS, 1600-546 Lisbon, Portugal; mafalda.aguiar@energypulsesystems.com (M.A.-M.); marcos.pereira@energypulsesystems.com (M.T.P.)
2   Pulsed Power Advanced Applications Group, Lisbon School of Engineering, GIAAPP/ISEL, 1959-007 Lisbon, Portugal
3   Carlos Silva Vinhos, Unip. Lda., 3500-728 Viseu, Portugal; carlos.wine@gmail.com
*   Correspondence: lmredondo@deea.isel.ipl.pt

**Abstract:** The increase in awareness of consumers and producers regarding the sustainable production and consumption of food commodities is motivating the emergence of new technologies to improve the efficiency of pre-established methods and reduce or supplant the usage of production factors. Thus, innovative technologies, such as the nonthermal application of pulsed electric fields (PEFs), may play a crucial role in the optimization of processes, both economically and environmentally (shrinkage of wastage, energy efficiency and decrease in the use of food additives), without compromising the quality of the final product. Thus, a comparison was made between the application of commercial-grade enzymes and PEF treatment to assess the impact on *cuvée* white grape must and on wine yield and quality. Oenological parameters were evaluated during alcoholic fermentation and after 3 months, with tartaric stability measured after 6 months. For this, assays resorting to 1.5 tons of Arinto grapes were separated into nine similar batches: three control, three treated with enzymes (1.5 g/100 kg) and three subjected to PEFs (1 kV/cm; 2 kJ/kg) at a rate of 4 ton/h. PEFs presented the highest increase in cuvée wine yield of 5.47%; a reduction of 19% of wine lees production was also determined. The effect of PEFs on pH, total acidity, turbidity, total phenols, color intensity, %Ye, total dry extract, volatile acidity and tartaric stability was studied and compared with control and enzymatic treatment. PEF and enzyme usage direct costs were determined; the employment of PEFs represented a direct cost of 0.12 EUR/ton, while enzyme usage was 1.80 EUR/ton.

**Keywords:** pulsed electric fields; white wine vinification; enzymes; yield; Arinto; PEF; tartaric stability

## 1. Introduction

The increasing awareness of producers and the general public regarding the production and consumption of sustainable food products [1,2], such as wine [3], is motivating the emergence of new processing technologies to improve the efficiency of pre-established methods and reduce or supplant the usage of production factors while assuring nutritional and organoleptic quality and food safety [4]. Thus, innovative technologies, such as the application of pulsed electric fields (PEFs), may play a crucial role in the optimization of processes, both economically and environmentally [5] (waste shrinkage, energy efficiency and decrease in the use of oenological products), representing a viable alternative to conventional methods that require the addition of organic solvents, longer extraction periods or additives, such as $SO_2$ [6], DMDC, or enzymes [7,8].

Despite the traditionalism associated with the wine sector, several technologies were introduced, accepted and widely disseminated. Commercially available enzymes, based on pectinases and many times associated with cellulases, hemicellulases, glucanases and proteases, perform as biocatalysts, contributing to and accelerating the degradation of the cell walls of the grape berries by hydrolysis of the polymers that are part of its constitution

(pectins, hemicelluloses and cellulose) [9]. This mechanism facilitates the liberation of the intracellular content by promoting the lysis of the cells and their organelles, such as vacuoles, contributing to the facilitation of mass transfer, thus optimizing juice yields, the extraction of compounds and reduction in viscosity and colloidal effects and having a positive effect in the clarification and filtration processes [9,10]. While enzymes are naturally present in grapes, they might have reduced concentrations or be inactivated due to other factors (e.g., presence of $SO_2$ and ethanol or low pH) [11,12]. Considered an additive, given its exogenous nature relative to the food matrix, enzymes used in the wine industry are mainly of microbial or fungal origin (e.g., *Lactobacillus*, *Aspergillus niger* and *Trichoderma* sp.) and can be applied to several stages of vinification [13]. However, the demand of both consumers and industry regarding the diminution of additive and preservative use has been pushing an agenda regarding the development of minimal intervention products. Thus, it is important to mention that, while the major risk of enzyme usage is currently associated with the inhalation of the product during its production and application, the residues left in wine are probably negligible, not posing a risk for consumers [10,14].

Over the last few years, the extensive application of pulsed electric fields in the food industry came under the spotlight considering its potential regarding microbial inactivation, leading to an enhancement of food safety, quality, stability and shelf-life and improving the phenomenon of mass transfer [15], together with the fact of it being an easily scalable technology capable of working in a continuous flow and short processing time [16,17] with low energetic requirements [18]. Furthermore, several researchers demonstrated that a PEF also displays the ability to increase yield during the pressing stage, without compromising the quality of the final product in several food matrixes, such as olive oil [19,20], other plant oils [21] and apple [22], tomato [23], citrus [24] and red fruit juice [25,26]. This allows one to establish a parallelism between the use of PEFs and enzymes in the food industry. Evidence of this occurrence in wine grape varieties, while existing, is quite scarce, not only given the typical variability amongst grape varieties and terroir but also mainly because most of these assays were performed at the lab or bench scale, highlighting the necessity of extrapolating the same results to a feasible industrial scale.

A PEF is considered a nonthermal technology, consisting of the exposition of biological material to pulsed electric fields capable of physically inducing the phenomenon of electroporation of the cell membranes.

One of the key advantages of using PEF in food processing is that it allows for the application of high-intensity electric fields without causing a significant increase in temperature. This is particularly important for the food industry, as it contributes to reducing energy consumption while preserving the nutritional and *flavour* characteristics of the food [27]. The variation in temperature during PEF treatment depends mainly on the specific energy input ($Ws$ = kJ/kg) used in the protocol [28,29]. This nonthermal effect of PEF is due to the short duration of the electrical pulses, usually in the microsecond (μs) range, and the brief duration of the treatment, typically lasting only a few milliseconds. The electrical energy is delivered in a disruptive way during PEF treatment, and it is not sufficient to cause significant heating of the food product, in contrast to traditional thermal processing methods [30]. Bearing this in mind, in this assay, the treatment applied does not significantly increase the temperature of the must ($\Delta T < 1\ °C$).

The approval of PEF as the pretreatment of musts by OIV was achieved in 2020, being included in the *Code of Oenological Practices* (Resolution OIV-OENO 634-2020) as the consequent corollary of extensive work realized by several authors, both at the laboratory and pilot-plant scale (e.g., FieldFood Project under Horizon 2020, grant no. 635632). However, a more industrial type of approach is needed, allowing the consolidation and proof-check of the feasibility and scalability needed in order to provide security and trust to industry peers.

Although the application of PEF as a pretreatment and its impact on wine characteristics (either sensorial or physico-chemical) on fermentation dynamics, yield and the final product have been considerably studied, studies mainly focused on red wine vinification

given the importance of phenolic substances on the quality parameters of red wine [7,16]. Thus, currently, there is a lacuna regarding the investigation of PEFs on white grape varieties, with a small number of publications regarding this topic [8,31–33], a few of them performed at a pilot-plant scale and even fewer comprising the assessment of the yield extraction optimization of *cuvée* (free run and low pressure, associated with high-quality wines) or total must volume. Furthermore, it is crucial to compare this innovative tool with other commonly used biotechnologies in winemaking, such as enzymes, to identify potential similarities and differences in achieving the same objectives. Additionally, our search of relevant databases and scientific journals revealed a lack of information regarding the volume of wine lees produced by using different methodologies to increase must yields. The aim of this study is to investigate the effects of pulsed electric field (PEF) treatment during the vinification of the Portuguese white grape variety Arinto at a pilot-plant scale, i.e., processing 2 T with a flow of 4 T/h. Specifically, we aim to compare the use of PEF treatment with the traditional use of enzymes during alcoholic fermentation and 3 months after vinification in terms of various parameters, including total phenols (TP); color intensity (CI); yellow (%Ye), red (%Re) and blue (%Bl) color components; the yield of *cuvée* juice, wine and lees; volatile acidity; turbidity; pH; total acidity; total dry extract; and tartaric stability (mini-contact test).

## 2. Materials and Methods

### 2.1. Grapes

Grapes (*Vitis vinífera* L.) of the Portuguese variety Arinto of Lisbon wine region provenance were harvested by hand at the optimum ripening stage and stored at 10 °C until their arrival at the pilot-plant winery located in Gouveia (Portugal). Upon arrival, the grape material was weighted and separated into similar batches before processing. The grapes were divided into 9 batches, each with about 167 kg, in which the only variation referred to the treatment applied: (a) 3 batches for control, (b) 3 batches treated with commercial-grade juice yield increasing enzymes and (c) 3 batches PEF-treated with monopolar-shaped pulses (1 kV/cm and 2 kJ/kg).

Testing PEF Parameters: Optimizing Preliminary Experiments

Prior to conducting the experiment at an industrial scale, preliminary assays were performed using a batch treatment chamber for the application of PEFs, using a linear parallel plate electrode (P2P: plate-to-plate) configuration, at a laboratory scale (EnergyPulse Systems, Lisbon, Portugal).

To accomplish this, batches of 200 g of destemmed grapes were weighed and crushed to simulate a typical vinification process and divided into six groups, each corresponding to a specific treatment type. Every experiment was carried out in triplicate. The range of parameters considered in each protocol of this preliminary experience was based not only on results published by other researchers but also taking into account the cost–benefit tradeoff between yield increase, equipment design requirements and energy efficiency [8,22].

Before undergoing PEF treatment, grapes were gently crushed, causing the skin to rupture, replicating the destemmer/crusher process. Afterwards, the PEF treatment was applied, and the grape mash went through a two-phase process. The first phase consisted of collecting the free run juice volume. In the second phase, the remaining grape mash was subjected to low pressure by being manually pressed using a ricer, representing the juice yield obtained by pressing. The optimization of *cuvée* yield being the principal focus of this work, only yield optimization was assessed a priori. The measure was performed with the aid of a graduated cylinder (±0.5 mL). The different PEF protocols and results obtained in this assay are presented in Table 1.

**Table 1.** Preliminary Results: Optimization of a PEF Protocol for Arinto Grapes.

|  | PEF Protocol | Free Run *Cuvée* Vol. (mL) | Free Run *Cuvée* Yield (%) | *Cuvée* Press Vol. (mL) | *Cuvée* Press Yield (%) | Total *Cuvée* Vol. (mL) | Total *Cuvée* Yield (%) |
|---|---|---|---|---|---|---|---|
| Control | - | $25.33 \pm 5.03$ | 12.7% | $20.00 \pm 4.0$ | 10.0% | $45.33 \pm 8.08$ | 22.7% |
| A | 1 kV/cm; 1 kJ/kg | $29.67 \pm 1.53$ | 14.8% | $23.67 \pm 3.21$ | 11.4% | $53.33 \pm 4.62$ | 26.2% |
| B | 1 kV/cm; 2 kJ/kg | $34.66 \pm 1.53$ | 17.3% | $18.67 \pm 0.58$ | 9.3% | $53.33 \pm 1.15$ | 26.6% |
| C | 1 kV/cm; 3 kJ/kg | $31.00 \pm 4.58$ | 15.5% | $21.66 \pm 3.21$ | 10.8% | $52.66 \pm 7.75$ | 26.3% |
| D | 0.5 kV/cm; 2 kJ/kg | $30.33 \pm 1.53$ | 15.2% | $18.00 \pm 2.65$ | 9.0% | $48.33 \pm 1.53$ | 24.2% |
| E | 1.5 kV/cm; 2 kJ/kg | $31.00 \pm 1.73$ | 15.5% | $16.67 \pm 2.08$ | 8.4% | $47.67 \pm 0.58$ | 23.9% |

Results presented as Mean ± Standard Deviation (n = 3).

Based on the obtained results, Protocol B was found to be the most effective PEF treatment, as it resulted in the highest increase in free run *cuvée* must (+36.8%) compared to the control, along with a 17.6% increase in total *cuvée* yield. It is noteworthy that the rise in free run *cuvée* yield was accompanied by a reduction in *cuvée* press yield (−6.6%). This suggests that PEFs may not only facilitate the extraction of greater volumes of must but also aid in obtaining higher-quality juices, a characteristic associated with *cuvée* musts.

*2.2. Vinification and PEF Application*

Immediately upon sorting, each batch was processed; see Figure 1. The first stage of vinification consisted of destemming and crushing the grapes, started by the unloading of the grape transport boxes to the destemmer/crusher (CMA Lugana 1), consisting of a destemming drum and two rubber crushing rollers. Commercial-grade pectolytic (1.5 g/100 kg) for juice yield increase was added at this point to the designated batches. $SO_2$ and ascorbic acid [34] were also added at this stage at a concentration of 15 mg/kg and 3 g/100 kg, respectively, to prevent oxidation in all batches. From there, the grape mash was pumped (Peristaltic Pump, CME PPC 200) at a rate of 4 ton/h through a continuous treatment co-axial chamber, where it was then subjected to PEF treatment in the required batches. To prevent disruption of the grape mash flow during pumping, a system of DN50 hoses was used.

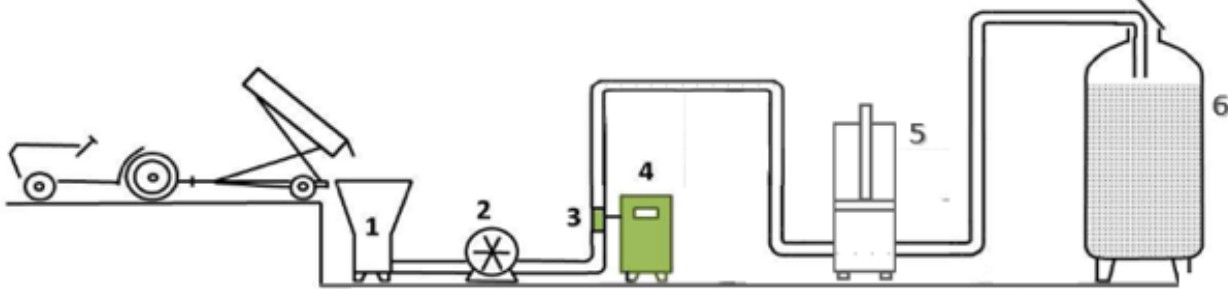

**Figure 1.** Simplified schematic of white grape vinification process, including the stage of PEF application: (1) Destemmer and crusher; (2) Peristaltic pump; (3) Co-linear PEF treatment chamber; (4) Pulse generator; (5) Press; and (6) Tank.

At the end of the line system, the grape mash would enter a pneumatic press (Bucher Vaslin® XPro 5, Chalonnes-sur-Loire, France) to start the next step of conventional white

wine vinification: the pressing stage. Considering that the main objective presented in this paper is to assess the possibility of a yield increase in *cuvée* must (originating from free run and 1st stage pressing) by using assisting technologies, enzymes, and pulsed electric fields. Therefore, the pressing cycle considered had a duration of 30 min, with a maximum applied pressure of 0.2 bar. At the end of pressing, each grape must was remitted to its inox tank, and commercial-grade pectolytic enzymes (1 g/hL) were added to all vats to aid the clarification of the musts before fermentation. Cold settling occurred during 48 h at a temperature of 9.5 °C ± 1. At the end of clarification, all musts were racked and inoculated with commercial yeast (Merit™, Chr Hansen, Denmark) to guarantee standardized microbiological fermentation conditions. Yeast nutrient was also added at a concentration of 30 g/hL. Alcoholic fermentation (AF) was conducted at 15 °C ± 1, and the temperature and density (Mustimetre Denis, Noizay, France) of the must were monitored every day until the end of fermentation. Sample collection occurred at five key stages of vinification previously decided:

1. Stage 0: immediately after pressing;
2. Stage 1: after racking;
3. Stage 2: in the middle of AF (around 1050 g/dm$^3$);
4. Stage 3: at the end of AF;
5. Stage 4: evolution after 3 months.

Exceptionally, samples were also collected 6 months after vinification, with the objective of assessing the tartaric stability of all wines under study.

For each parameter analysis, three replicates were made. For stabilization, free SO$_2$ was corrected to 38 mg/L at the end of AF.

The PEF system used was constituted by a high-voltage solid-state Marx generator (EPULSUS® BM3B-15, EnergyPulse Systems, Lisbon, Portugal), with 15 kV/400 A and 9 kW average power, capable of delivering almost perfectly square monopolar- and bipolar-shaped pulses, and a DN50 co-linear treatment chamber with a diameter of 50 mm and a distance between electrodes of $d$ = 5 cm; see Figure 2.

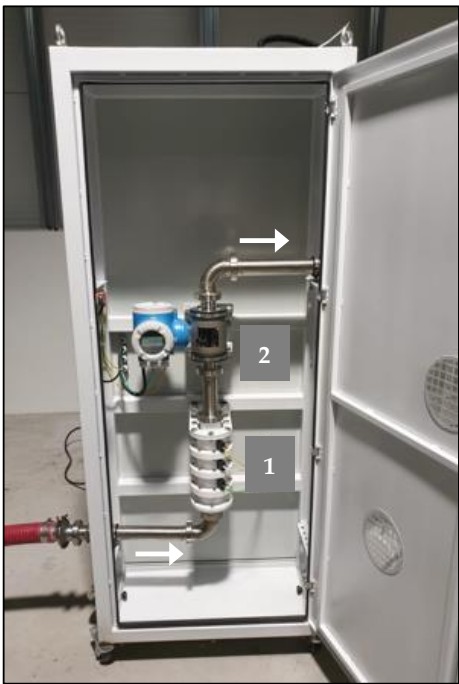

**Figure 2.** PEF equipment picture, including the DN50 treatment chamber (1), flowmeter (2) and the flow of must.

Considering the most relevant parameters regarding PEF treatments, the electric field amplitude *E* (kV/cm) and specific energy *Ws* (kJ/kg), the following relations apply:

$$E = \frac{U}{d} \tag{1}$$

with *U* being the pulse voltage amplitude applied in kV and *d*, in cm, being the distance between the electrodes of the co-linear chamber, and

$$W_s = \frac{W_t}{m} \tag{2}$$

with $W_t$ being the total applied energy, in J, and *m*, as the mass of the material treated, being, in this specific case, grape must in kg. $W_t$ is obtained through

$$W_t = UIt_{on}N \tag{3}$$

where *I*, in A, is the pulse current amplitude, $t_{on}$ is the pulse width in μs, and *N* is the number of applied pulses [19]. Furthermore, it is also possible to access the thermal impact of the PEF treatment by calculating the temperature variation, in °C, using the following equation:

$$\Delta T = \frac{W_s}{C_p} \tag{4}$$

where $C_p$ is the specific heat capacity of the material under treatment, which is ~4.3 J.kg$^{-1}$.K$^{-1}$ for wine and musts [19,35]. Regarding this information, it is easy to access the theoretical $\Delta T$, obtaining, for this case, an increase in temperature of 0.47 °C at the end of the treatment chamber, which has an irrelevant impact on the winemaking process and quality.

### 2.3. Must, Wine and Wine Lees Yield

One of the objectives of this work is to determine the impact of PEF on grape juice yield and, therefore, assess the added value due to an increase in production while also diminishing the volume of byproducts with a very low energy input due to the pulsed nature of the energy applied. The grapes were weighed and separated before vinification, with the aid of an electronic weighing scale ADAM GBC 60 (Adam Equipment Ltd., Milton Keynes, UK). The volume of must or wine was calculated using a graduated height gauge to measure the height of the liquid in the cylindrical tank, allowing for the determination of the corresponding volume of must or wine obtained.

The calculation of grape juice yield and wine lees was determined by the percentage ratio between the volume obtained and the yield of the grapes before processing [22], being calculated as follows:

$$Yield = \frac{V_f}{m_t}100 \tag{5}$$

where $V_f$ is the volume obtained of juice after pressing or wine lees and $m_t$ is the total grape mass weighed. The volume of total wine lees comprises the sum of the results measured in two different stages of vinification: after the cold juice settling with the aid of pectolytic enzymes and after the end of AF.

The yield of the wine obtained after vinification was calculated as

$$Must\ Yield - Wine\ lees = Final\ Wine\ Yield \tag{6}$$

determining the final yield of the *cuvée* wine produced.

### 2.4. pH and Total Acidity

The pH of the must and wine samples was determined by a SesION + pH31 benchtop pH meter with an integrated magnetic stirrer (Hach, Loveland, CO, USA).

Total acidity was monitored using method OIV-MA-AS313-01, through titration using bromothymol blue (4 g/L) and 0.1 mol/L Sodium hydroxide (Vinilab, Amadora, Portugal). The results were expressed in grams of tartaric acid per liter.

### 2.5. Spectrophotometry: Total Phenols (TP), Color Intensity (CI) and Color Components (%Ye, %Re and %Bl)

All assays comprising the utilization of spectrophotometry techniques were realized at *Comissão Vitivinicola da Região do Dão*'s official laboratory. Previous to all analytical assessments, all of the samples were centrifuged at 5000 rpm for 10 min. All of the assays were performed using spectrophotometry (U-2900 Spectrophotometer, Hitachi, Tokyo, Japan).

Total phenol (TP) content was assessed by an internal protocol (MI22—Revisão 4, 2020) where, after pretreatment, the absorbance of the sample was read at 280 nm in quartz cuvettes with a 10 mm optical path length.

The determination of the color parameters was carried out through the application of the Glories method, presented at the *Compendium of International Methods of Analysis* by OIV (OIV-MA-AS2-07B:R2009). Thus, the absorbances of the samples were determined at 420, 520 and 620 nm, resorting to a quartz cell with optical path $b$ of 1 cm. According to Glories [36,37], color intensity (CI) can be calculated as

$$CI = \frac{A420 + A520 + A620}{b} \tag{7}$$

Furthermore, it is also possible to assess the contribution of compounds of the color yellow (%Ye), red (%Re) and blue (%Bl) from

$$Ye = \frac{A420}{CI} \times 100 \ Re = \frac{A520}{CI} \times 100 \ Bl = \frac{A620}{CI} \times 100 \tag{8}$$

The absorbance of wine at A420 is utilized by several authors as a simple index for the determination of browning of the wines to relate with the oxidation of the product [38,39]. In our case, we will use the %Ye color compound as an indicator of browning.

### 2.6. Turbidity

Turbidity (*T*) was assessed to determine the concentration of suspended particles and clarity of must/wine with the utilization of a 2100 Q portable turbidimeter (Hach, Loveland, CO, USA) and is expressed in nephelometric turbidity units (NTU). Turbidity was assessed in Stage 0 (immediately after pressing), Stage 1 (after decantation), Stage 3 (end of AF) and Stage 4 (after 3 months).

### 2.7. Total Dry Extract

The total dry extract (TDE) was determined according to the procedure established by OIV at the *Compendium of International Methods of Analysis* (Method OIV-MA-AS2-03B). It is also known as total dry matter and measures the totality of compounds present in the wine of non-volatile characterization (such as sugars, acids and minerals) and is represented in g/L. Considering that wine is an organic matrix, the total dry extract is constituted of both organic and inorganic compounds; thus, the choice of different winemaking techniques is expected to affect this parameter [40].

### 2.8. Volatile Acidity

Volatile acidity (VA) consists of the measurement of volatile acids present in the matrix, mainly constituted by acetic acid but also including other volatile acids, such as lactic, formic and butyric acid [41]. Every wine has volatile acidity and, as long as the aromatic



threshold is not reached, it normally remains between 0.6 and 0.9 g/L, depending on the style of wine and consumer's sensibility [42]. For this paper, the protocol used was performed according to the procedure published by OIV (Method OIV-MA-AS313-02).

### 2.9. Tartaric Stability

To assess the tartaric stability of the wines 6 months after vinification, the protocol used was based on the *DIT* test (*degree d'instabililité tartrique*) adapted by Bosso et al. (2016) [43]. This test consists of the continuous measure of the conductivity of a sample, for 4 min, at 0°, prior to and after the addition of micronized potassium bitartrate crystals, inducing secondary crystallization. The conductivity drop ($\Delta\chi\%$) is the indicator of wine tartaric stability. For instance, white wines with $\Delta\chi\% > 5$ are considered stable. The threshold for rosé or light-bodied red wines is 4%, while the advised $\Delta\chi\%$ is > 3 for full-bodied, polyphenolic-rich, red wines [43–45].

### 2.10. Sampling, Results Presentation and Statistical Analysis

For every tank of each batch, analyses were performed in triplicate, with these, as previously mentioned, being collected over 5 stages of vinification. Collected data for each parameter are represented as mean ± standard deviation in the respective tables.

Graphs presented in this paper are a visual characterization of mean and standard deviation; each data point represents the mean value of each replicate tank (e.g., ARC1, ARC2 and ARC3). Column graphs present superscripted letters assigned to each column. Similar letters indicate that the means are not significantly different among them, whereas different letters indicate statistically significant differences in the mean values.

The collected data were analyzed using IBM SPSS Statistics, Version 28.0.1.0 (SPSS Inc., Chicago, IL, USA), with a statistical significance level of $\alpha = 0.05$ calculated for each factor. When necessary, post hoc testing between groups of samples for each parameter was assessed using Tukey's test, allowing for the determination of differences in the independent variable. Eta-squared ($\eta^2$) is a measure of effect size associated with the parametric analysis of variance (ANOVA); higher values indicate a greater proportion of differences between subjects that can be attributed to the factor under study. For example, if an $\eta^2$ value of 0.648 is presented, it can be interpreted as the treatment factor being responsible for 64.8% of the assessed differences.

The error from measuring the height of the liquid in the cylindrical tank was calculated as ±0.5 L, about 1% of the yields measured.

## 3. Results and Discussion

### 3.1. Fermentation Kinetics

As previously mentioned, after processing, the grapes were subjected to a temperature of 9.5 °C ± 1 for 48 h to promote cold settling; after racking, alcoholic fermentation developed at controlled temperatures of 15 °C ± 1. Alcoholic fermentation was terminated 25 days after the arrival of the grapes, when the density presented values <995 g/dm³ and the content of D-Glucose+D-Fructose was <1 g/L. No differences in the fermentation dynamics were determined between subjects.

This result is in concordance with the conclusions of Comuzzo et al. 2007, who presented similar results while working with white grape variety Garganega at an average fermenting temperature of 25 °C [33]. Other authors also reported no alterations in terms of fermentative dynamics in red grape varieties (Cabernet Sauvignon, Merlot, Tempranillo, Garnacha and Graciano e Mazuelo) [37,46,47].

### 3.2. Must, Wine and Wine Lees Yield
Must and Wine

The resulting yield obtained for the three protocols under study is represented in Table 2.

**Table 2.** Preliminary Results: Optimization of a PEF Protocol for Arinto Grapes, where ARC represents the control vinification samples, ARE the ones treated with enzymes and ARPEF the samples subjected to PEF.

|  | Treatment Protocol | *Cuvée* Must Yield (%) | Total Lees Yield(%) | *Cuvée* Wine Yield (%) |
|---|---|---|---|---|
| ARC | - | 53.41 ± 1.17% | 14.12 ± 0.38% | 39.29 ± 1.76% |
| ARE | 1.5 g/100 kg | 57.17 ± 1.37% | 18.23 ± 1.23% | 38.94 ± 4.09% |
| ARPEF | 1 kV/cm; 2 kJ/kg | 56.21 ± 1.13% | 14.77 ± 1.20% | 41.44 ± 1.87% |

Results presented as Mean ± Standard Deviation (n = 3).

One-Way ANOVA was conducted to compare the mean scores of *Cuvée* must yield obtained across three different groups. The results indicated that there was no significant difference between the groups; $F_{(2, 6)} = 3.519$, $p = 0.097$, $\eta^2 = 0.54$. The volume of wine lees produced was also assessed in two stages, immediately after must clarification and after the end of alcoholic vinification, considered to calculate the final yield of the *cuvée* wine produced. While the differences in *cuvée* wine yield obtained after the vinification process did not reach statistical significance ($F_{(2, 6)} = 1.617$, $p = 0.274$, $\eta^2 = 0.324$), it is still relevant to compare the mean values of the final cuvée wine yield. ARPEF achieved the highest mean yield (41.44%), suggesting that the application of PEF may result in a yield increase of +5.47%. In practical terms, this means that for every 100 kg of grapes, the producer can now obtain an additional 5.4 L of *cuvée* wine. Surprisingly, while enzymes presented the highest percentage of volume extracted immediately after pressing, the final yield of *cuvée* wine was the lowest, mainly due to the higher production of wine lees, combined with the highest variation in results obtained with enzymes (ARE).

Fauster et al. 2020 also reported that no significant results were achieved when comparing the application of pectolytic enzymes for yield increase, with a combination of PEF and enzymes, on a white grape blend constituted by Grüner Veltliner and Traminer varieties [8]. However, several studies presented different conclusions. Praporscic et al., 2007 performed on three white grape varieties (Muscadelle, Sauvignon and Semillon) and demonstrated an increase in total yield from 49–54% to 76–78% after applying a PEF protocol of 0.75 kV/cm and W ≈ 20 kJ/kg [31]. Grimi et al. 2009 reported similar results (67% to 75%) in total yield for the variety Chardonnay when subjected to a PEF treatment of 0.4 kV/cm and W ≈ 15 kJ/kg [48]. These contrasting conclusions may not only be attributed to the variations in the treatments applied to each grape mash but also to the inherent variability observed within the groups. Working with biological materials, such as grapes, can be challenging due to the difficulty in achieving complete homogeneity in the sample. Even when sharing a *terroir*, the variability of collected grape material cannot be completely overlooked, as individuals exhibit plasticity in response to their environment. This is explained by Bradshaw (1965) as the extent to which environmental factors influence the expression of individual traits in a genotype [49]. The unique combination of factors, such as light exposure, temperature [50], spatial soil variation among plant vines (including differences in hydric and nutritional conditions) and agricultural practices, exerts influence on each berry, resulting in varying compositions, even within the same cluster [51].

Total wine lees yield was obtained with the sum of wine lees measured over two different stages: after juice settling and after the end of AF. The results obtained translated in the highest yield of lees produced by the batch treated with pectolytic enzymes, 18.23 ± 2.13%, with control and PEF-assisted batches demonstrating approximate values of, respectively, 14.11 ± 0.65% and 14.78 ± 2.08%. After performing statistical analysis, it is concluded that,

while not statistically significant, it was close to being considered significant ($F_{(2, 6)} = 3.093$, $p = 0.058$, $\eta^2 = 0.612$), indicating a large effect size of the type of treatment. A comparison between ARPEF and ARE suggests that the production of this byproduct could experience a reduction of 18.9%.

To the best of our knowledge, there is currently no information available on the effect caused by PEF on wine lees production. However, it is important to note that wine lees, which are the sediments that settle at the bottom of wine barrels after fermentation, are believed to account for 25% of the total waste byproducts produced by wineries [52]. Therefore, the use of PEF technology has the potential to significatively and positively impact the management of and reduction in wine lees waste generated during wine production.

### 3.3. pH and Total Acidity

The obtained oenological parameters over the five stages of all of the wines under study are presented in Table 3 and Figure 3.

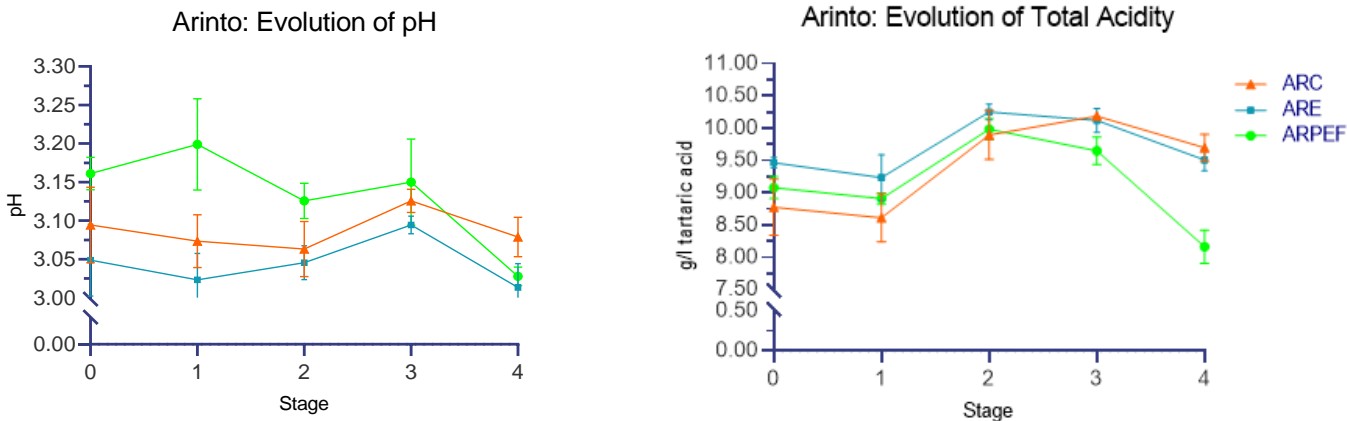

**Figure 3.** pH (**left**) and Total Acidity (**right**) evolution over time (Stage 0: after pressing, Stage 1: after decantation, Stage 2: middle of AF, Stage 3: End of AF and Stage 4: after 3 months) (n = 9).

Upon statistical analysis, the type of treatment applied showed a significant effect on both pH ($F_{(2, 24)} = 26.675$, $p < 0.001$, $\eta^2 = 0.690$) and total acidity (*TA*) ($F_{(2, 24)} = 22.161$, $p < 0.001$, $\eta^2 = 0.648$).

The post hoc Tukey's HSD test was used to perform multiple comparisons for both pH and TA. All groups presented a statistically significant difference in pH regarding treatment type (ARC and ARE: $p = 0.007$, 95% C.I. = 0.0103, 0.702; ARC and ARPEF: $p = 0.002$, 95% C.I. = $-0.0773$, $-0.0174$; and ARE and ARPEF: $p < 0.001$, 95% CI = $-0.1175$, $-0.0576$). The same was observed for TA (ARC and ARE: $p = 0.008$, 95% C.I. = $-0.4893$, $-0.0698$; ARC and ARPEF: $p = 0.008$, 95% C.I. = 0.0698, 0.4893; and ARE and ARPEF: $p = <0.001$, 95% C.I. = 0.3494, 0.7689).

Regarding Stage 0, the lowest pH values were observed in ARE ($3.05 \pm 0.02$), while the highest were seen in ARPEF ($3.16 \pm 0.01$); ARC had the lowest TA value at $8.77 \pm 0.44$ g/L of tartaric acid. However, 3 months after vinification, ARC exhibited the highest pH and TA at $3.07 \pm 0.025$ and $9.70 \pm 0.21$ g/L of tartaric acid, respectively. Meanwhile, ARPEF and ARE had pH values of $3.03 \pm 0.012$ and $3.01 \pm 0.03$, and TA values of $8.16 \pm 0.25$ and $9.51 \pm 0.17$ g/L of tartaric acid, respectively.

**Table 3.** pH and Total Acidity Parameter Results.

| | pH | | | | | Total Acidity * | | | | |
|---|---|---|---|---|---|---|---|---|---|---|
| Stage | 0 | 1 | 2 | 3 | 4 | 0 | 1 | 2 | 3 | 4 |
| ARC1 | 3.06 ± 0.01 | 3.10 ± 0.02 | 3.03 ± 0.01 | 3.11 ± 0.01 | 3.09 ± 0.01 | 8.50 ± 0.09 | 8.40 ± 0.08 | 9.39 ± 0.02 | 10.19 ± 0.02 | 9.50 ± 0.05 |
| ARC2 | 3.16 ± 0.03 | 3.09 ± 0.01 | 3.11 ± 0.00 | 3.14 ± 0.00 | 3.10 ± 0.01 | 8.47 ± 0.03 | 8.33 ± 0.00 | 10.13 ± 0.00 | 10.19 ± 0.06 | 9.63 ± 0.09 |
| ARC3 | 3.07 ± 0.01 | 3.03 ± 0.01 | 3.05 ± 0.01 | 3.05 ± 0.01 | 3.05 ± 0.01 | 9.35 ± 0.09 | 9.10 ± 0.04 | 10.16 ± 0.04 | 10.16 ± 0.04 | 9.95 ± 0.05 |
| ARE1 | 3.04 ± 0.01 | 3.04 ± 0.01 | 3.02 ± 0.00 | 3.08 ± 0.00 | 2.99 ± 0.01 | 9.70 ± 0.02 | 9.29 ± 0.08 | 10.33 ± 0.04 | 10.29 ± 0.02 | 9.60 ± 0.13 |
| ARE2 | 3.01 ± 0.03 | 2.99 ± 0.04 | 3.05 ± 0.01 | 3.10 ± 0.00 | 2.99 ± 0.01 | 9.53 ± 0.00 | 9.58 ± 0.22 | 10.33 ± 0.04 | 10.19 ± 0.02 | 9.63 ± 0.05 |
| ARE3 | 3.10 ± 0.02 | 3.04 ± 0.03 | 3.07 ± 0.00 | 3.10 ± 0.01 | 3.05 ± 0.01 | 9.36 ± 0.05 | 8.84 ± 0.15 | 10.09 ± 0.04 | 9.88 ± 0.04 | 9.30 ± 0.08 |
| ARPEF1 | 3.14 ± 0.02 | 3.12 ± 0.00 | 3.13 ± 0.02 | 3.12 ± 0.00 | 3.04 ± 0.01 | 9.23 ± 0.13 | 8.97 ± 0.06 | 10.19 ± 0.02 | 9.86 ± 0.04 | 7.83 ± 0.05 |
| ARPEF2 | 3.16 ± 0.01 | 3.24 ± 0.01 | 3.10 ± 0.00 | 3.11 ± 0.02 | 3.03 ± 0.01 | 8.93 ± 0.13 | 8.94 ± 0.06 | 9.90 ± 0.07 | 9.70 ± 0.04 | 8.30 ± 0.05 |
| ARPEF3 | 3.18 ± 0.02 | 3.24 ± 0.00 | 3.15 ± 0.00 | 3.22 ± 0.01 | 3.01 ± 0.01 | 9.08 ± 0.13 | 8.81 ± 0.04 | 9.85 ± 0.04 | 9.38 ± 0.05 | 8.35 ± 0.04 |

* Results presented in g/L of tartaric acid (n = 3).

Our pH results are consistent with those obtained by López-Alfaro et al. (2013), who subjected grapes of the red varieties Graciano, Tempranillo and Grenache to a 4–6 kV/cm electrical field and observed a significant increase in pH [53]. However, the results were contradictory regarding TA, as the end of AF showed that ARPEF had a lower concentration (9.65 ± 0.21 g/L of tartaric acid). In a study performed by López et al. (2008) on Tempranillo, electrical fields of 5–10 kV/cm were applied, and no significant differences were observed at the end of AF [37]. Similarly, Comuzzo et al. 2008 reported no significant differences, although they noted a slight increase in pH values, which was undetectable after fifty days of AF. He also hypothesized that the increase in pH was due to the salification process of organic acids, which resulted in the release of hydroxide ions ($OH^-$) [33].

### 3.4. Total Phenols (TP), Color Intensity (CI) and Color Components (%Ye and %Re)

In white wine, the principal compounds responsible for coloration are catechins and hydroxycinnamates, which, initially, might present colorless characteristics but can form yellow or brown products when enzymatically (mainly during AF) or nonenzymatically oxidized [38,54]. This process is denominated *browning*, and it is not desired in white wine production, with a few exceptions (sherries and sweet fortified wines, like Port and Madeira) [34]. Several authors use $A_{420}$ as an indicator of the browning rate, which naturally increases over time due to the oxidation and polymerization of phenolic compounds; however, it is important to mention that browning depends not only on the grape variety, maturity at the time of harvest and technological processes but also on temperature, pH, light and oxygen content, which affect the oxidation rate of wines as a whole [38,39].

The increase in polyphenolic extraction of red wines due to PEF pretreatment is already well-known, being a major advantage for the utilization of this technology [16,55,56]. In white wine, the increase in extraction might present a duality of advantages, such as the increase in juice yield and aromatic compounds, and disadvantages, such as higher bitterness and astringency, and accelerate aging; however, it is dependent on parameters such as grape variety and maceration time [8,32,57]. Ideally, the solution is equilibrating the PEF protocol and skin contact period, if used.

To better understand the impact of PEF and assess differences in the dynamics of polyphenol content during AF, we measured total polyphenols (TP), color intensity (CI), %Ye, %Red and %Blue. We also compared these measurements between a control group and batches treated with extraction enzymes. The results are presented in Table 4 and resumed in Figure 4.

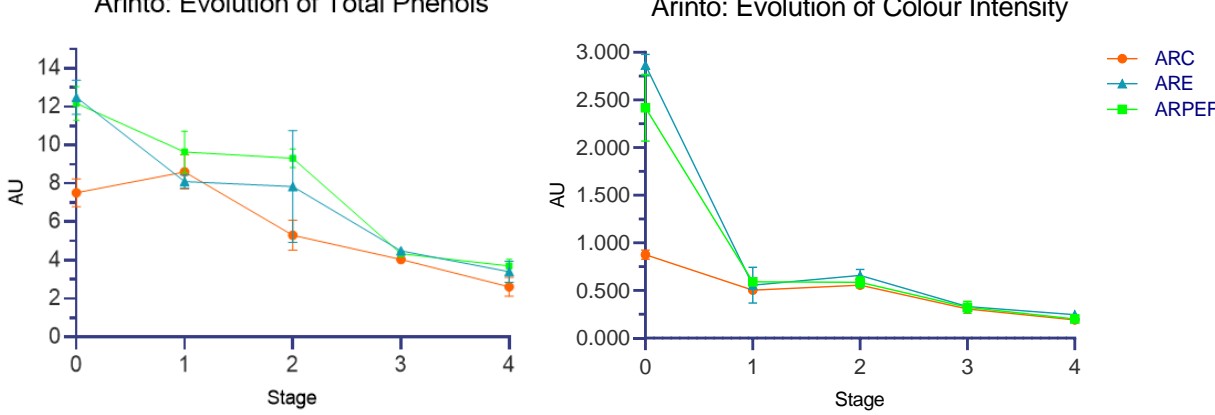

**Figure 4.** Evolution of Total Phenols (TP) (**left**) and Color Intensity (**right**) over time (Stage 0: after pressing, Stage 1: after decantation, Stage 2: middle of AF, Stage 3: End of AF and Stage 4: after 3 months) (n = 9).

**Table 4.** Total Phenols and Color Intensity Parameter Results.

| | Total Phenols (*TP*) | | | | | Color Intensity (*CI*) | | | | |
|---|---|---|---|---|---|---|---|---|---|---|
| Stage | 0 | 1 | 2 | 3 | 4 | 0 | 1 | 2 | 3 | 4 |
| ARC1 | $7.4 \pm 0.01$ | $7.5 \pm 0.06$ | $5.1 \pm 0.06$ | $3.9 \pm 0.08$ | $2.0 \pm 0.02$ | $2.808 \pm 0.02$ | $0.465 \pm 0.00$ | $0.540 \pm 0.00$ | $0.317 \pm 0.00$ | $0.190 \pm 0.00$ |
| ARC2 | $6.8 \pm 0.03$ | $9.5 \pm 0.08$ | $4.6 \pm 0.05$ | $4.3 \pm 0.09$ | $2.8 \pm 0.03$ | $2.778 \pm 0.02$ | $0.504 \pm 0.00$ | $0.543 \pm 0.00$ | $0.291 \pm 0.00$ | $0.186 \pm 0.00$ |
| ARC3 | $8.4 \pm 0.08$ | $8.9 \pm 0.05$ | $6.3 \pm 0.04$ | $4.0 \pm 0.02$ | $3.1 \pm 0.02$ | $3.016 \pm 0.0$ | $0.544 \pm 0.01$ | $0.585 \pm 0.00$ | $0.311 \pm 0.00$ | $0.199 \pm 0.00$ |
| ARE1 | $11.3 \pm 0.04$ | $7.7 \pm 0.05$ | $5.4 \pm 0.06$ | $4.7 \pm 0.05$ | $3.1 \pm 0.01$ | $0.928 \pm 0.02$ | $0.423 \pm 0.00$ | $0.660 \pm 0.00$ | $0.310 \pm 0.00$ | $0.214 \pm 0.00$ |
| ARE2 | $13.3 \pm 0.02$ | $8.0 \pm 0.05$ | $6.5 \pm 0.10$ | $4.4 \pm 0.03$ | $3.0 \pm 0.01$ | $0.878 \pm 0.02$ | $0.808 \pm 0.01$ | $0.729 \pm 0.00$ | $0.345 \pm 0.00$ | $0.278 \pm 0.00$ |
| ARE3 | $12.9 \pm 0.03$ | $8.5 \pm 0.09$ | $11.7 \pm 0.07$ | $4.3 \pm 0.08$ | $4.1 \pm 0.07$ | $0.823 \pm 0.01$ | $0.442 \pm 0.01$ | $0.583 \pm 0.01$ | $0.337 \pm 0.00$ | $0.247 \pm 0.00$ |
| ARPEF1 | $11.2 \pm 0.02$ | $8.2 \pm 0.08$ | $9.0 \pm 0.06$ | $4.4 \pm 0.04$ | $3.4 \pm 0.04$ | $1.993 \pm 0.01$ | $0.629 \pm 0.00$ | $0.645 \pm 0.00$ | $0.407 \pm 0.00$ | $0.227 \pm 0.00$ |
| ARPEF2 | $13.2 \pm 0.03$ | $10.1 \pm 0.11$ | $10.0 \pm 0.08$ | $4.4 \pm 0.07$ | $3.6 \pm 0.05$ | $2.802 \pm 0.02$ | $0.552 \pm 0.01$ | $0.598 \pm 0.01$ | $0.289 \pm 0.00$ | $0.211 \pm 0.00$ |
| ARPEF3 | $12.1 \pm 0.02$ | $10.6 \pm 0.11$ | $9.0 \pm 0.09$ | $4.2 \pm 0.05$ | $4.2 \pm 0.05$ | $2.458 \pm 0.02$ | $0.593 \pm 0.01$ | $0.515 \pm 0.00$ | $0.277 \pm 0.00$ | $0.162 \pm 0.00$ |

Results presented in absorbance units (AU). n = 3.

Similar to the results observed in pH and TA, total phenols (TP) was also significantly affected by the wine treatment used. Based on the statistical results (F(2, 24) = 33.941, $p < 0.001$, $\eta^2 = 0.738$), we can hypothesize that 73.8% of the variance in total phenols concentration can be attributed to differences related to the processing methods. To locate which groups presented statistical differences amongst them, Tukey's test of multiple comparisons was performed. ARC and ARPEF presented the largest differences amongst the three subjects, with a mean difference (MD) of $-2.220$ ($p < 0.001$, 95% C.I. = $-2.919$, $-1.521$), followed by ARC and ARE, with MD = $-1.647$ ($p < 0.001$, 95% C.I. = $-2.345$, $-0.948$). However, when comparing the groups ARPEF with ARE, the results do not show statistically significant differences, with the MD = 0.573 ($p = 0.122$. C.I. = $-0.125$, 1.272).

The type of treatment used during grape processing, control, enzymes and PEF, also presented significant differences in the color intensity (F(2, 24) = 308.008, $p < 0.001$). Post hoc Tukey's test demonstrated that the highest mean difference was regarding ARC vs. ARE (MD= $-0.445$, $p < 0.001$. 95% C.I. = $-0.492$, $-0.398$). ARE vs. ARPEF posed a smaller difference that was still statistically significant (MD= 0.108, $p < 0.001$, 95% C.I. = 0.061, 0.155). However, when analyzing the color components %Ye, %Red and %Blue separately, we noticed significant differences between treatments (%Red: F(2, 24) = 19.128, $p < 0.001$, $\eta^2 = 0.614$; %Ye: F(2, 24) = 85.665, $p < 0.001$, $\eta^2 = 0.877$; and %Blue: F(2, 24) = 36.936, $p < 0.001$, $\eta^2 = 0.754$). Post hoc tests revealed significant differences between ARC and both ARE and ARPEF for all color components, but no statistically significant differences were found between ARE and ARPEF.

Comuzzo et al., 2008 observed similar results regarding color intensity and total phenols for control vs. PEF-treated white wine [33].

As previously mentioned, %Ye is used in this experiment as a browning indicator. Analyzing the results shows that the main differences were observed in Stages 0 and 4, where ARC had a significantly higher %Ye compared to both ARE and PEF. This may indicate that the use of PEF or enzymes could increase wine resistance to oxidation. However, it is also important to note that, although not statistically significant, ARE had a lower concentration of compounds contributing to yellow coloration than PEF. One possible hypothesis to explain these results is related to the extent of membrane destruction and/or pore size. It is possible that greater membrane destruction, such as the presence of larger pores, may lead to the release of more complex and polymerized polyphenolic molecules, which are larger in size and therefore unable to pass through the cell membranes of the control grape mash. These molecules might contribute to better stability of the phenolic fraction, therefore leading to a reduction in the oxidation processes, and as a result, the reduction in the browning reaction [33].

### 3.5. Turbidity

The obtained oenological parameters over the four stipulated stages of all of the wines under study are presented in Table 5 and resumed in Figure 5.

The Kruskal–Wallis test was conducted to examine the turbidity difference between all treatments in Stages 0, 1, 3 and 4. Statistically significant differences were achieved in Stage 0 (H = 9.951, $p = 0.007$), most specifically between ARC and ARPEF ($p = 0.011$) batches and ARC vs. ARE pairs ($p = 0.004$). ARE and ARPEF presented a not statistically significant difference ($p = 0.744$). It is worth mentioning that ARC presented the lowest turbidity immediately after processing, with a value of $1792 \pm 213.8$NTU, followed by ARE and ARPEF with similar values of, respectively, $2959.7 \pm 271$NTU and $2693 \pm 77.97$NTU.

After cold settling and first racking (Stage 1), at the end of alcoholic fermentation and 3 months post-AF, significant differences were also determined (H = 11.153, $p = 0.004$; H = 10.27, $p = 0.006$; H = 16.613, $p < 0.001$, respectively). Computing pairwise comparisons resulted in conclusions similar to the ones obtained for Stage 0.

**Table 5.** Results of the parameter Turbidity.

| Stage | Turbidity | | | |
|---|---|---|---|---|
| | 0 | 1 | 3 | 4 |
| ARC1 | 945 ± 2.1 | 501 ± 2.3 | 528 ± 0.3 | 16 ± 0.3 |
| ARC2 | 2135 ± 58.3 | 308 ± 3.5 | 443 ± 0.6 | 92 ± 0.2 |
| ARC3 | 2299 ± 72.0 | 444 ± 2.9 | 358 ± 2.1 | 8 ± 0.3 |
| ARE1 | 3557 ± 33.3 | 322 ± 1.5 | 185 ± 0.6 | 3 ± 0.0 |
| ARE2 | 1875 ± 31.2 | 130 ± 0.6 | 263 ± 10.1 | 2 ± 0.1 |
| ARE3 | 3447 ± 43.7 | 157 ± 1.0 | 406 ± 1.2 | 6 ± 0.0 |
| ARPEF1 | 2415 ± 164.6 | 214 ± 3.1 | 10 ± 0.1 | 8 ± 0.1 |
| ARPEF2 | 2872 ± 105.0 | 537 ± 4.7 | 438 ± 4.0 | 2 ± 0.0 |
| ARPEF3 | 2790 ± 74.4 | 763 ± 6.1 | 355 ± 2.1 | 1 ± 0.1 |

Results presented in Nephelometric Turbidity Units (NTU). n = 3.

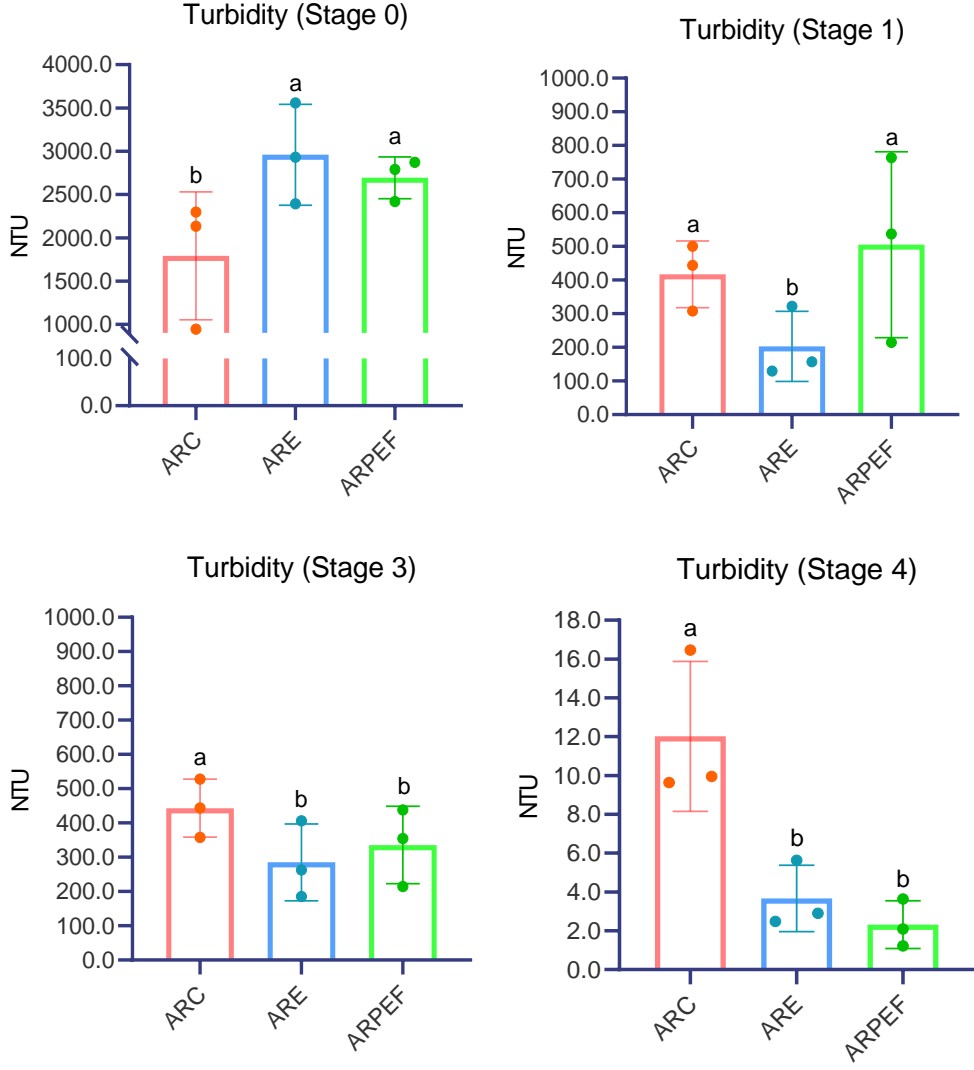

**Figure 5.** Results of turbidity (NTU) analyzed on 4 stages of vinification (Stage 0: after pressing (top left graph), Stage 1: after decantation (top right graph), Stage 3: End of AF (bottom left graph) and Stage 4: after 3 months (bottom right graph)) (n = 9). Statistically significant differences ($p < 0.05$) are indicated by different letters assigned to the means.

While Comuzzo et al., 2018 obtained similar results to our experiment [33], Praporscic et al., 2007 found the opposite after assessing *T*, demonstrating that an increase in grape juice yield was accompanied by a decrease in *T*. This was attributed to possible selective damage to the cell membrane, promoting its degradation to a lesser extent [31]. These differences may be due to variations in grape varieties, winemaking techniques and equipment (e.g., type of press) [33]. However, when experimenting on Chardonnay grapes, the results obtained by Grimi et al. 2009 showed that *T* was not affected by PEF pretreatment in various pressing conditions [48]. Studies performed on other matrixes, such as apple and carrot juice, corroborate the results presented by Praporscic et al. 2007 [58,59]. Differences among protocols used in the experiments, such as contact with the solid fraction, pressure exerted or other variables, might be responsible for the variability of the results obtained; therefore, more experiments must be conducted in order to fully understand the relation between the use of PEF and *T*.

### 3.6. Total Dry Extract

Three months after vinification, total dry extract (TDE) was assessed, and it was concluded that ARE presented the highest concentration, 23.47 ± 0.07, while ARC and ARPEF had, respectively, 22.2 ± 0.25 and 22.72 ± 0.37 g/L, presenting statistically significant differences (H= 6.950, *p* = 0.031) (Figure 6). Pairwise comparisons revealed a significant difference between the ARC and ARE wine groups (*p* = 0.009) but no significant differences between the ARC and ARPEF (*p* = 0.359) or ARPEF and ARE (*p* = 0.092).

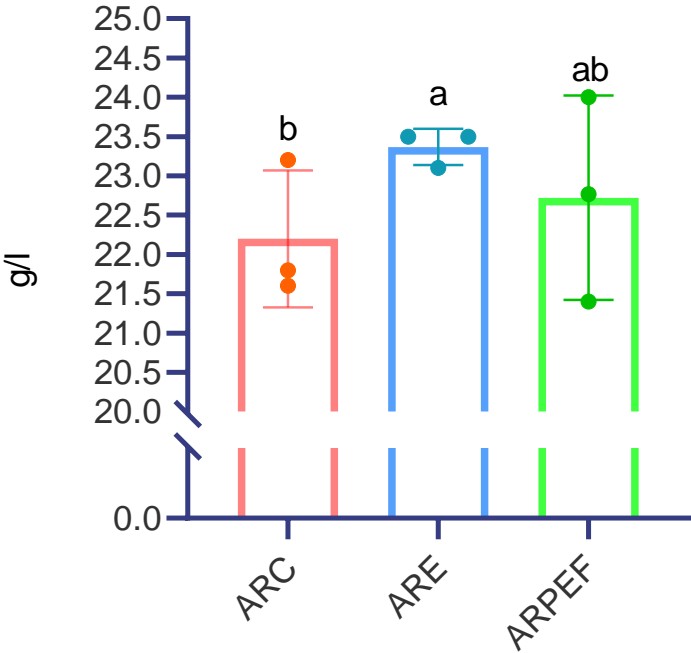

**Figure 6.** Total Dry Extract results 3 months after the end of AF (Stage 4) (n = 9). Statistically significant differences (*p* < 0.05) are indicated by different letters assigned to the means.

This is in accordance with Comuzzo et al., 2018, who also reported a slight increase in TDE when working with the white grape variety Garganega. As expected, PEF demonstrated its ability to optimize the extraction of intracellular compounds [60], possibly due to its facilitation of the extraction of minerals from vegetal tissues, as demonstrated in assays realized with alfalfa juice [61] and phenolic compounds from grape skins [15,33,37].

### 3.7. Volatile Acidity

Volatile acidity, mainly comprising acetic acid, is regulated by current legislation, with the maximum value for this parameter, for commercialized wines, being 1.2 g/L. The concentration of finished wines varies, being approximately 0.5 g/L. Acetic acid is mainly produced by the *Acetobacter* species, and, to a lesser extent, by yeasts during alcoholic fermentation, being considered a spoilage characteristic in excessive quantities [62]. Its synthesis depends on several factors, such as grape sanity, techniques and processes, stage of vinification, species and strains of microorganisms present in the medium, exposure to oxygen and nutrient imbalance, which lead to competition between existent yeast and bacterial populations [63,64].

The results obtained for this parameter are discriminated in Figure 7, and statistical analysis using the Kruskal–Wallis test showed no significative differences between control, enzymatic treatment and PEF (H= 0.226, $p$ = 0.893).

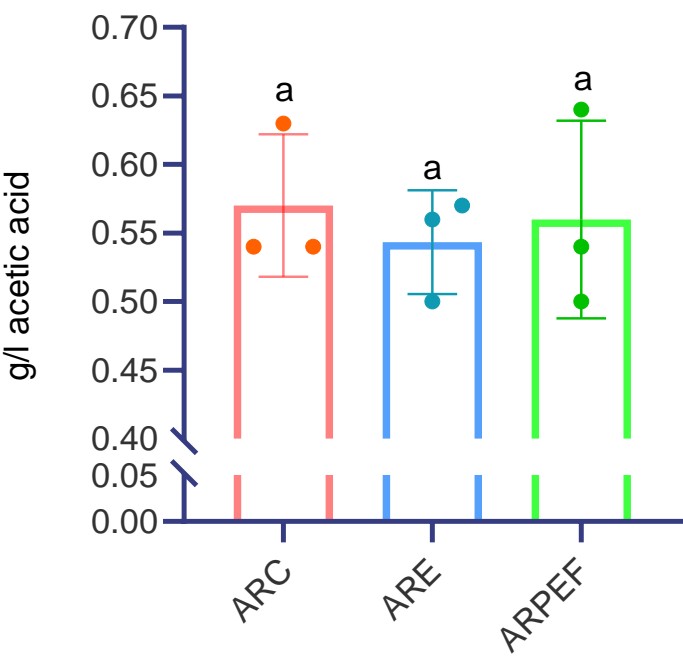

**Figure 7.** Results of Volatile Acidity 3 months after the end of AF (Stage 4) (n = 9). Statistically significant differences ($p < 0.05$) are indicated by different letters assigned to the means.

Given the lack of published information on PEF application in white grape must, we were forced to compare our results with the existing data for both red and white musts. Recalling that the PEF protocol used for this assay was 1 kV/cm and 2 kJ/kg, the information is in favor of what was concluded by Comuzzo et al., 2018, when applying a PEF protocol of 1.5 kV/cm and 22 kJ/kg to white variety Garganega grapes. However, when Comuzzo applied the same electrical field strength and lowered the specific energy to 11 kJ/kg, the difference was considered significantly higher (Tukey HSD test, $p < 0.05$) [33]. Puértolas et al. performed a comparison between enzyme and PEF (5 kV/cm; 3.67 kJ/kg) application to study the effects on the maceration of Cabernet Sauvignon must, concluding that neither PEF nor enzymes usage interfered with this parameter [7].

### 3.8. Tartaric Stability

Tartaric acid is one of the main organic acids present in grapes, must and wine [40]. However, its supersaturated presence can lead to one of the most common physico-chemical instability problems in enology: tartrate precipitation [44]. This leads to the appearance of

crystal sediments, not only during alcoholic fermentation but also over the course of the aging and storage period [44,45]. While being a natural phenomenon, its appearance is considered a quality fault and is mainly observed in white wines, considering the usual lower temperatures of conservation [44]. Therefore, the assessment of the tartaric stability of wines and possible posterior stabilization treatment are strongly advised. The most common stabilization method relies on temperature effects (cold stabilization), ion removal (electrodialysis and ion-exchange resins) or introducing stabilization additives, such as mannoproteins, metatartaric acid and carboxymethyl cellulose (CMC) [65–67]. These treatments represent considerable costs for the wineries. Tartaric acid (H2T) dissociates and reacts with $Ca^{2+}$, forming precipitates of Calcium Tartrate (CaT), or with $K^+$, creating potassium hydrogen tartrate (KHT) [67]. The crystallization kinetics can be affected by temperature, wine composition, colloidal matter, pH, alcohol content, KHT initial supersaturation and, of course, the concentration of cations, such as $Ca^{2+}$ and $K^+$ [44,67,68]. Considering that potassium might be stored in the vacuole of berries' skin cells [69], the use of processes that may affect the structural integrity of lipidic membranes, such as PEF and enzymes [70], can promote the extraction of cations and alter the dynamic of this natural phenomenon. Thus, it was decided that tartaric stability would be assessed 6 months after vinification, allowing to a better understanding of the possible effects of each biotechnology in this matter. The obtained results are presented in Table 6 and Figure 8.

**Table 6.** *Mini-Contact* test Results.

| | $\Delta$**Conductivity ($\Delta\chi$%)** | $\Delta$**Conductivity ($\Delta\chi$%)** |
|---|---|---|
| ARC1 | $9.33 \pm 0.68$ | |
| ARC2 | $7.37 \pm 0.89$ | $8.48 \pm 1.23$ |
| ARC3 | $8.73 \pm 1.33$ | |
| ARE1 | $5.56 \pm 0.63$ | |
| ARE2 | $4.81 \pm 1.10$ | $5.51 \pm 0.87$ |
| ARE3 | $6.16 \pm 0.15$ | |
| ARPEF1 | $4.94 \pm 0.67$ | |
| ARPEF2 | $2.85 \pm 0.18$ | $3.01 \pm 1.74$ |
| ARPEF3 | $1.24 \pm 1.17$ | |

Results presented in $\Delta\mu$S/cm. n = 3.

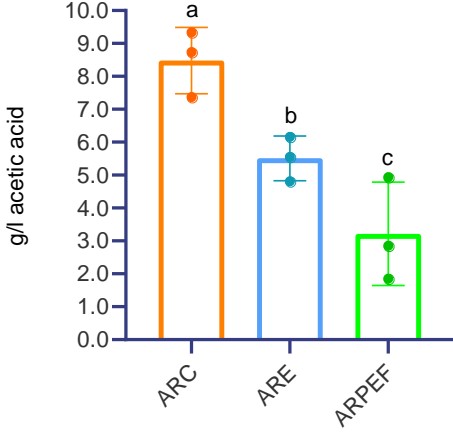

**Figure 8.** Tartaric Stability assessed 6 months after the end of AF (Stage 4) (n = 9). Statistically significant differences ($p < 0.05$) are indicated by different letters assigned to the means.

Statistical analysis was performed using one-way ANOVA, which demonstrated not only significative differences between all treatments but also a large effect size of approximately 78.5% ($F_{(2, 24)} = 26.675$, $p < 0.001$, $\eta^2 = 0.785$). The obtained results prove the possibility of using both PEF and enzymes to contribute to a higher level of natural stability of white wines. However, considering that only PEF-treated wines present a drop in conductivity of $\Delta\chi\% > 5\%$, as stipulated by Bosso et al., (2016), only this group is considered stable from the tartaric point of view [43].

A comprehensive review of the literature revealed, to the best of our knowledge, no published articles regarding the effect of PEF application on the tartaric stability of wines. However, based on (a) the results obtained, (b) PEF and enzymes action on bilipidic membranes [71], (c) the high presence of $K^+$ in the vacuoles [69] and (d) the mechanism responsible for the precipitation of tartrates described by Lasanta and Goméz, 2012 [67], we can hypothesize that a higher release of those salts in the initial stages promotes a faster salification process of tartrates, which leads to achieving earlier natural stabilization.

*3.9. PEF Economics*

Given the present results, PEF can increase the added value of white grape vinification, in particular Arinto vinification, either by juice yield increase or by increasing product quality and therefore value and allowing the reduction in production factors, indirectly contributing also to cost reduction.

Direct costs, as described by Lasanta and Gomes, 2012 [67], include energy and water consumption, chemicals, labor, consumables and wine losses. Given the inherent design of the PEF installation in the winery, particularly its inline configuration and little maintenance needs, there are no requirements for process modifications: thus, labor and water consumption alterations can be disregarded, as well as consumables. Concerning wine losses, as previously mentioned in this paper, PEF-treated wines presented lower lees production, very close to being statistically significant ($p = 0.058$). This suggests that PEF treatment can potentially reduce wine loss directly while also potentially contributing to an increase in cuvée wine yield. For comparison purposes, we can examine the direct costs, which are virtually only associated with energy consumption. For instance, in the present assay, with an applied $Ws = 2$ kJ/kg at a rate of 4 ton/h, this process requires an energetic input of 2.22 kW per hour. Considering an average energy cost of 0.22 EUR/kWh, the application of this protocol represented a cost of 0.49 EUR/h and 0.12 EUR/ton. In contrast, the cost of the commercial-grade enzyme, recommended at a minimum dosage of 15 g/ton, amounts to 1.80 EUR/ton of processed grapes. Consequently, a direct cost saving of 94% can be observed.

In addition, it is important to note that PEF treatment demonstrated a positive effect in the tartaric stability of the wine under study; thus, it is possible to hypothesize that it can also indirectly promote the economic sustainability of the product by avoiding the need to employ other techniques, such as the usage of stabilizing additives that prevent the formation of crystals, such as mannoproteins (direct costs (DC) = 3.0 EUR/hL), metatartaric acid (DC = 0.07 EUR/hL) and Carboxymethylcellulose (CMC) (DC = 0.7 EUR/h), or processes such as cold stabilization (DC = 0.76–3.74 EUR/hL), electrodialysis (DC = 0.56–3.1 EUR/hL) and ion exchange (0.07 EUR/hL) [67,72].

**4. Conclusions**

Although there is a growing interest in the application of PEF in the wine industry for microbial inactivation and optimizing mass transfer, including extraction yield and phenolic extraction, there is still a significant gap in knowledge regarding the effects of PEF on white grape varieties and the resulting impact on wine quality.

One of the most interesting applications of PEF is optimizing *cuvée* must and wine yield, which are often associated with wines with a higher-quality profile. Thus, this coupled with the potential to increase the total wine yield suggests that PEF could have a positive economic and sustainability impact. However, both the lack of peer-reviewed studies

and assays performed at the pilot or industrial scale contribute to a lacuna of information, which acts as an obstacle to a wider acceptance of this technique in the industry. With this experiment, we hope to contribute to a better understanding of PEF as a scalable, economic, sustainable and easy-to-incorporate technology. For this, the comparison of PEF with other yield-improving and extraction techniques already well established in the industry, such as the case of yield extraction enzymes, is essential.

With a protocol of 1 kV/cm and 2 kJ/kg, PEF was able to increase the *cuvée* wine yield to 5.47%, while the use of yield extraction enzymes demonstrated a surprising reduction of −0.89% when compared with control. This difference is partially justified by the reduction in the production of byproducts; while PEF produced +4.6% lees compared to control, it was able to reduce its production by −18.95% when compared with the enzymatic treatment.

Significant differences were found over the course of AF and 3 months after vinification for pH and TA between all subjects. TP, IC and %*Red*, %*Ye* and %*Bl* demonstrated significant differences between control and both treatments, However, significant differences were not found between the application of enzymes and PEF. Using %*Ye* as a browning indicator, we found that control had a significantly higher possible susceptibility to oxidation, which allows us to conclude that the use of PEF and extraction enzymes might contribute to the increase in antioxidative properties, which can be justified by the grade of polymerization of the polyphenolic molecules extracted.

Turbidity was significantly distinct between control and both ARPEF and ARE, where both treatments presented higher NTUs in the first stages of vinification. However, after 3 months, ARPEF presented the lowest turbidity, followed by ARE, when compared to control.

Regarding the total dry extract, assessed 3 months after vinification, it was determined that the enzymatic treatment presented a concentration significantly higher than ARC. PEF presented no differences between both groups; however, it was considerably lower when compared to ARE. Volatile acidity was also not affected significantly between all groups after 3 months.

Among the possible novel findings of this study, in addition to the unexpected reduction of lees during vinification, another notable result was the rapid achievement of tartaric stability through PEF-assisted vinification.

Based not only on our results but also on those of other authors, the application of pulsed electric fields (PEFs) in the winemaking industry holds great promise and merits further research to better understand and unlock its full potential.

**Author Contributions:** Conceptualization, M.A.-M., L.M.R., M.T.P. and C.S.; methodology, M.A.-M. and L.M.R.; formal analysis, M.A.-M., L.M.R., M.T.P. and C.S.; investigation, M.A.-M., L.M.R., M.T.P. and M.A.-M.; writing—review and editing, M.A.-M. and L.M.R.; project administration, M.T.P. All authors have read and agreed to the published version of the manuscript.

**Funding:** This research was developed under the project "PureWine—Increasing quality and production capacity of European Wine Industry through an innovative Pulsed Electric Field based process applied to vinification", grants CENTRO-01-0247-FEDER-041392 and LISBOA-01-0247-FEDER-041392.

**Institutional Review Board Statement:** Not applicable.

**Informed Consent Statement:** Not applicable.

**Data Availability Statement:** Data are available if necessary.

**Conflicts of Interest:** EnergyPulse Systems develops and commercializes PEF equipment that can be used for wine vinification.

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
