# Peer review of "Pulsed Electric Fields vs. Pectolytic Enzymes in Arinto Vinification: Effects on Yield and Oenological Parameters"

_applsci, doi:10.3390/app13148343_

Round 1

Reviewer 1 Report

Dear authors,

The manuscript deals with an important and interesting topic related to applying Pulsed Electric Fields (PEF) in the wine industry. Based on experimental data obtained from the experiment, the authors concluded that applying Pulsed Electric Fields (PEF) in the winemaking industry holds great promise and merits further research to understand better and unlock its full potential.   

The technical quality of the manuscript is remarkable in terms of how the experimental results are presented and related to the scientific data obtained by other researchers. Also, the scientific quality of the manuscript does rise to the scientific level of Applied Science. There are no further comments, and the manuscript can be published in Applied Science.

Author Response

Dear authors,

The manuscript deals with an important and interesting topic related to applying Pulsed Electric Fields (PEF) in the wine industry. Based on experimental data obtained from the experiment, the authors concluded that applying Pulsed Electric Fields (PEF) in the winemaking industry holds great promise and merits further research to understand better and unlock its full potential.    

The technical quality of the manuscript is remarkable in terms of how the experimental results are presented and related to the scientific data obtained by other researchers. Also, the scientific quality of the manuscript does rise to the scientific level of Applied Science. There are no further comments, and the manuscript can be published in Applied Science.

Reply: Authors thank the reviewer for the encouraging remarks.

Reviewer 2 Report

Positive comments to the authors:

1. The topic of the manuscript is current, as it concerns the application of new ecological non-enzymatic methods, namely Pulsed Electric Fields instead of Pectolytic Enzymes in Vinification and a study of effects on Yield and Oenological Parameters;

2. The introduction shows the level of science in the studied area and the prospects for the development of the research problem;

3. The materials and methods used in the scientific research are presented in relative detail;

4. The obtained results are illustrated with the help of tables and figures;

5. The Results and Discussion section shows the erudition of the authors and their ability to dig deep into the bowels of science. In the presented manuscript, the authors attempt to explain the obtained scientific results and indicate the direction of future research in the field of wine biochemistry and the introduction of new non-enzymatic physical methods of impact;

6. At the end of the manuscript, valuable conclusions and recommendations are presented for the widespread use of Pulsed Electric Fields for grape must liquefaction instead of expensive enzymatic depectinization in the vinification process.

Negative notes and recommendations:

1. Please zoom in figures 1, 3, 5, 6, 7 and 8 by 10-15% for better visualization;

2. I recommend refining the text and removing some inaccuracies - for example, on line 40, the formula for sulfur dioxide - SO2 to SO2 etc.

3. I recommend final editing of the text by an English-speaking editor for final polishing of the English;

4. You could create an abbreviations subsection in which to systematize all abbreviations;

5. I recommend continuing research with other grape varieties for the production of red and rosé wine;

I recommend final editing of the text by an English-speaking editor for final polishing of the English;

Author Response

Positive comments to the authors:

  1. The topic of the manuscript is current, as it concerns the application of new ecological non-enzymatic methods, namely Pulsed Electric Fields instead of Pectolytic Enzymes in Vinification and a study of effects on Yield and Oenological Parameters;
  2. The introduction shows the level of science in the studied area and the prospects for the development of the research problem;
  3. The materials and methods used in the scientific research are presented in relative detail;
  4. The obtained results are illustrated with the help of tables and figures;
  5. The Results and Discussion section shows the erudition of the authors and their ability to dig deep into the bowels of science. In the presented manuscript, the authors attempt to explain the obtained scientific results and indicate the direction of future research in the field of wine biochemistry and the introduction of new non-enzymatic physical methods of impact;
  6. At the end of the manuscript, valuable conclusions and recommendations are presented for the widespread use of Pulsed Electric Fields for grape must liquefaction instead of expensive enzymatic depectinization in the vinification process.

Reply: Authors thank the reviewer for its comments and recommendations, which helped to increase the quality of the paper.

All the changes are highlighted in yellow.

Negative notes and recommendations:

  1. Please zoom in figures 1, 3, 5, 6, 7 and 8 by 10-15% for better visualization;

Reply: The figures in question were zoomed, according to the recommendation proposed by the reviewer.

  1. I recommend refining the text and removing some inaccuracies - for example, on line 40, the formula for sulfur dioxide - SO2 to SOetc.

Reply: All of the text was reviewed, being the inaccuracies corrected, namely the SO2 formula, on the lines 40, 52, 167, 201, and the substitution of all of “p.ex.” for the correct abbreviation “e.g.”, line 52, 54 and 541.

 Other inaccuracies regarding language were also altered: substitution of “to” by “with” on line 41, “everyday” altered to “every day” in line 192; “.” Altered to “,” on line 323.

 Alteration of the treatment protocol displayed on the table 2, from “1g/100kg” to “1.5g/100kg” (line 343); The chemical formula “OH-“ was corrected to “OH-“ on line 433.

On line 681 “associated to wines” was substituted by “associated with wines”. On the line 704, “on” was altered to “in”

  1. I recommend final editing of the text by an English-speaking editor for final polishing of the English;

Reply: the text was reviewed.

  1. You could create an abbreviations subsection in which to systematize all abbreviations;

Reply: In fact, there were some abbreviations that were not identified clearly. This was added in the Table 2:

“where ARC represent the control vinification samples, ARE the ones treated with enzymes and ARPEF the samples subjected to PEF.”

  1. I recommend continuing research with other grape varieties for the production of red and rosé wine;

Reply: We will continue the research on this filed.

Reviewer 3 Report

Pulsed Electric Fields vs Pectolytic Enzymes in Arinto Vinification: effects on Yield and Oenological Parameters

Overall, the subject of the research is interesting. The introduction is based on the current literature and the Authors explained the purposefulness of the research.

I have the following comments on this manuscript. The following points should be taken into account:

LINE 32 p.1 Please check publication citation requirements. Quoting in square brackets should be appropriate. Please standardize the citation throughout the text, so that it is the same (especially in chapter 3 Results and Discussion - do not give the date of publication in round brackets – lines 358, 362, 364, 372, 421, 425, 427, 490, 533, 539, 541, 584, 642, 648)

LINE 120 p.3 “provenient” correct the spelling (provenience or provenance– this word will be correct

LINE 321 p.8 Remove spaces between the ‘1’ and the ‘%’ the same in line 662/p.17

LINE 326 p.7 Insert degree sign ‘°’ generally used as in line 190/p5 and line 187

LINE 341 p.9 Remove the double % sign from the value ‘41.44 ± 1.87’ in Table 2

LINE 397 p.10 Table 3 - I would recommend rounding the standard deviation values to two decimal places for better readability of the results in Table 3, the same applies to Table 4 (line 457)

LINE 738 p.18 Please note the requirements of the Journal regarding References, there is a comma after the author's name, and after the letter of the name there is a dot “.” and “;” . The year of publication should be in bold.

Author Response

Overall, the subject of the research is interesting. The introduction is based on the current literature and the Authors explained the purposefulness of the research.

I have the following comments on this manuscript. The following points should be taken into account:

Reply: Authors thank the reviewer for its comments and recommendations, which helped to increase the quality of the paper.

All the changes are highlighted in yellow.

LINE 32 p.1 Please check publication citation requirements. Quoting in square brackets should be appropriate. Please standardize the citation throughout the text, so that it is the same (especially in chapter 3 Results and Discussion - do not give the date of publication in round brackets – lines 358, 362, 364, 372, 421, 425, 427, 490, 533, 539, 541, 584, 642, 648)

Reply: Changes were performed on lines 360, 363, 366, 430, 493, 536, 537, 542, 544, 557, 586, and all of the citations were altered to follow MDPI citation requirements, with the square brackets.

LINE 120 p.3 “provenient” correct the spelling (provenience or provenance– this word will be correct

Reply: The sentence was changed to “ Arinto of Lisbon wine region provenance” on line 120.

LINE 321 p.8 Remove spaces between the ‘1’ and the ‘%’ the same in line 662/p.17

Reply: Change made.

LINE 326 p.7 Insert degree sign ‘°’ generally used as in line 190/p5 and line 187

Reply: All of the text was reviewed, with several degree signs being altered, namely on lines 89, 121, 216, 219, 328. 329, and 335.

LINE 341 p.9 Remove the double % sign from the value ‘41.44 ± 1.87’ in Table 2

Reply: Change made.

LINE 397 p.10 Table 3 - I would recommend rounding the standard deviation values to two decimal places for better readability of the results in Table 3, the same applies to Table 4 (line 457)

Reply: Change made.

LINE 738 p.18 Please note the requirements of the Journal regarding References, there is a comma after the author's name, and after the letter of the name there is a dot “.” and “;” . The year of publication should be in bold.

Reply: All of the References were reviewed and changed to fit the requirements of the Journal.

Reviewer 4 Report

The manuscript showed an applicable study that using PEF for Arinto Vinification which is worth to publish. However, some question may needed to clarify. First of all, one-way ANOVA and tukey's test was used in the results. It is suggested to display significant different groups with related symbol. In table 2, ARPEF were compared with ARE and control groups. IN line 349, it stated that there was no significant difference between groups of Cuvee Wine Yield. However, it stated that ARPEF was the highest yield in line 352. The standard deviation make the results seems very close of ARPEF and ARE groups. Is it still can claim that ARPEF has the highest values? In the results of table 3, 4 and 5, it is unstable of the three trials in stage 4. For example, the total acidity ranged from 7.83 to 8.35 g/L of tartaric acid, Colour intensity ranged from 0.162 to 0.227 and turbidity ranged from 1 to 8. What are the reasons behind and can it be improved?Finally, PEF economics of PEF was investigated in 3.4. Did the equipment and establishment cost were counted in the calculation? If these costs were involved, is it still can assess the saving of 94%?

Author Response

The manuscript showed an applicable study that using PEF for Arinto Vinification which is worth to publish. However, some question may needed to clarify.

Reply: Authors thank the reviewer for its comments and recommendations, which helped to increase the quality of the paper.

All the changes are highlighted in yellow.

First of all, one-way ANOVA and tukey's test was used in the results. It is suggested to display significant different groups with related symbol.

Reply: The superscripted letters were added to column charts (line 524, 566, 585 and 642. Also, information regarding the subscripted letters was added to the column graph legends and on the lines 312, 313 and 314.

In table 2, ARPEF were compared with ARE and control groups. IN line 349, it stated that there was no significant difference between groups of Cuvee Wine Yield. However, it stated that ARPEF was the highest yield in line 352. The standard deviation make the results seems very close of ARPEF and ARE groups. Is it still can claim that ARPEF has the highest values?  

Reply: The sentence was rewritten to improve clarity regarding the data considered, line 355:

“While the differences in Cuvée Wine yield obtained after the vinification process did not reach statistical significance (F(2, 6) = 1.617, p = 0.274, η²=0.324), it is still relevant to compare the mean values of the final cuvée wine yield. ARPEF achieved the highest mean yield (41.44%), suggesting that the application of PEF may result in a yield increase of +5.47%. In practical terms, this means that for every 100kg of grapes, the producer can now obtain an additional 5.4 liters of cuvée wine.”

In the results of table 3, 4 and 5, it is unstable of the three trials in stage 4. For example, the total acidity ranged from 7.83 to 8.35 g/L of tartaric acid, Colour intensity ranged from 0.162 to 0.227 and turbidity ranged from 1 to 8. What are the reasons behind and can it be improved?

Reply: This variation in the data may be attributed to a combination of several factors, which are challenging to control from a practical standpoint. These factors primarily involve the inherent variability within each tank, taking into account convection movements and potential heterogeneity despite homogenization and proper sample collection. Furthermore, to a lesser extent, we can associate the variation with method errors, operator errors, and possible minor variations related to the material used (grapes). It is important to acknowledge that even within the same cluster, differences among berries can arise due to various factors.

Finally, PEF economics of PEF was investigated in 3.4. Did the equipment and establishment cost were counted in the calculation?. If these costs were involved, is it still can assess the saving of 94%?

Reply: The equipment and establishment cost were unaccounted considering that those are regarded as indirect costs However, to avoid misconceptions, alterations were performed to explain what type of costs are considered as direct costs, line 667.          

“Direct costs, as described by Lasanta and Gomes, 2012  [67], include energy and water consumption, chemicals, labor, consumables, and wine losses. Given the inherent design of the PEF installation in the winery, particularly its inline configuration and little maintenance needs, there are no requirements for process modifications: thus, labour and water consumption alterations can be disregarded, as well as consumables. Concerning wine losses, as previously mentioned in this paper, PEF treated wines presented lower lee production, very close to being statistically significative (p = 0.058). This suggests that PEF treatment can potentially reduce wine loss directly, while also potentially contributing to an increase in cuvée wine yield. For comparison purposes, we can examine the direct costs, which are virtually only associated with the energy consumption. For instance, in the present assay, with an applied Ws=2 kJ/kg at a rate of 4 ton/h, this process requires an energetic input of 2.22 kW per hour. Considering an average energy cost of 0.22 €/kWh, the application of this protocol represented a cost of 0.49 €/h and 0.12 €/ton. In contrast, the cost of the commercial-grade enzyme, recommended at a minimum dosage of 15 g/ton, amounts to 1.80 €/ton of processed grapes. Consequently, a direct cost saving of 94% can be observed.”